# Does Shiftwork Impact Cognitive Performance? Findings from the Canadian Longitudinal Study on Aging (CLSA)

**DOI:** 10.3390/ijerph191610124

**Published:** 2022-08-16

**Authors:** Rea Alonzo, Kelly K. Anderson, Rebecca Rodrigues, Neil Klar, Paolo Chiodini, Manuel Montero-Odasso, Saverio Stranges

**Affiliations:** 1Department of Epidemiology & Biostatistics, Schulich School of Medicine & Dentistry, Western University, London, ON N6A 5C1, Canada; 2Lawson Health Research Institute, London, ON N6C 2R5, Canada; 3Department of Psychiatry, Schulich School of Medicine & Dentistry, Western University, London, ON N6A 5C1, Canada; 4Medical Statistics Unit, University of Campania “Luigi Vanvitelli”, 80138 Naples, Italy; 5Gait and Brain Laboratory, Parkwood Institute, Lawson Health Research Institute, London, ON N6C 0A7, Canada; 6Department of Medicine, Division of Geriatric Medicine, Schulich School of Medicine & Dentistry, Western University, London, ON N6A 5C1, Canada; 7Departments of Family Medicine and Medicine, Schulich School of Medicine & Dentistry, Western University, London, ON N6A 5C1, Canada; 8Department of Precision Health, Luxembourg Institute of Health, Strassen, L-1445 Luxembourg, Luxembourg

**Keywords:** shiftwork, shift schedules, psychological distress, cognitive performance, CLSA

## Abstract

Few large nationwide studies have investigated the relationship between shiftwork and cognitive performance, and little is known about whether and how psychological distress may impact this relationship. This study aimed to examine: (1) the cross-sectional relationship between shiftwork (yes/no) and some aspects of cognitive performance (declarative memory and executive functioning) and (2) the potential moderating effect of psychological distress among 20,610 community-dwelling adults from the comprehensive cohort of the Canadian Longitudinal Study on Aging (CLSA). Differences by sex and retirement status were also explored. Shiftwork was significantly associated with poorer performance for executive functioning (interference condition: ß = 0.47, 95% CI: 0.31 to 0.63; MAT: ß = −0.85, 95% CI: −1.21 to −0.50) but not for declarative memory. Completely and not/partly retired males showed poorer cognitive performance on executive functioning. However, no evidence of a moderating effect by psychological distress was found. Our findings confirm the association between shiftwork and cognitive performance and highlight important health correlates of shiftwork.

## 1. Introduction

Shiftwork is defined as non-standard work hours occurring in the hours before 7:00 a.m. or after 6:00 p.m. [1], and has become more prevalent worldwide [2], with specific industries and occupations highly dependent on the shift system [1,3]. Shiftwork is associated with sleep loss [4] and the development of numerous health conditions which may be a product of disrupted circadian rhythms due to abnormal work hours [5]. As cognitive processes are regulated by the endogenous circadian clock [6,7], shiftwork may impair cognitive functioning as well [8]. Shiftwork has also been found to share common risk factors for cognitive impairment, including higher risks of social isolation, being overweight, and unhealthy lifestyles including smoking [9]. As such, shiftwork could be a potential modifiable risk factor that may have implications for the development of cognitive impairment. 

Potential acute and chronic effects of shiftwork on cognitive function have been examined by relatively few population-based studies, and even fewer Canadian studies have examined this topic. For example, studies assessing the short-term effects of shiftwork on cognition using French and Swedish population-based data suggest an association between shiftwork and lower cognitive performance for processing speed and executive functioning [10,11]. Chronic effects of shiftwork examined by a few population-based studies, as well as data from the Nurses’ Health Study, have generated conflicting findings [8,11,12,13]. One Canadian study by Wong and colleagues [6] used a cross-sectional sample of 4255 Canadian workers and found significant effects of work stress and sleep quality on the relationship between shiftwork and subjective cognitive function.

Given the nature of shiftwork, shift schedules can also negatively impact an individual’s mental health [14,15]. In Europe and North America, 15% to 20% of people in the workforce were affected by psychological distress [16,17], which has been identified as an associated risk factor for cognitive impairment [18]. Therefore, investigating the potential moderating role of psychological distress may help elucidate the association between shiftwork and cognitive performance. 

Currently, there are very few nationally representative Canadian studies which have investigated the relationship between shiftwork and cognitive performance using objective measures. The potential impact of mental health on this relationship has not been previously examined. Thus, the objectives of this study were to: (1) examine the association between shiftwork and cognitive performance and (2) explore whether psychological distress might moderate the relationship between shiftwork and cognitive performance. As males and females exhibit differences in health outcomes [6], we explored whether differences in sex existed in this relationship. We also investigated differences in retirement status as it has been found to be associated with cognitive impairment [19].

## 2. Materials and Method

### 2.1. Data Source and Study Setting

Data from the Canadian Longitudinal Study on Aging (CLSA), a national longitudinal study which collects information from a random stratified sample of Canadians aged 45–85 years old at enrollment [20], were obtained for this study. The CLSA excluded people residing in the Northwest Territories, Yukon, and Nunavut, federal First Nations reserves, provincial First Nations settlements, as well as institutionalized individuals and full-time members of the Canadian Armed Forces [20]. Baseline data from 51,338 participants were collected between 2010 and 2015 [20]. The CLSA sample was divided into two cohorts: a Tracking cohort and a Comprehensive cohort [20]. Participants who participated in telephone interviews consisted of the Tracking cohort (N = 21,241) [20]. The Comprehensive cohort (N = 30,097) included participants who completed in-person home interviews and lived within 25 to 50 km of the data collection sites, located in seven of the provinces [20].

We used cross-sectional data from the CLSA Comprehensive cohort due to the availability of the cognitive performance variables in this sample. People at each level of retirement (completely, partly, or not retired) were included in the study. Only participants considered full-time workers at either their main or only job [21] among the partly and not retired groups, were included in our study. We excluded participants considered to be part-time workers as well as participants with missing data on employment and retirement status. Participants who had never worked or who were currently unemployed at the time of the survey were also excluded. Data access and research ethics board approval were respectively obtained from the CLSA (Application Number: 190247) and the Western University Health Science Research Ethics Board (Project Number: 112140). 

### 2.2. Shiftwork Variables

Participants who self-reported being partly or not retired were asked to describe their work schedules for the activity they considered their main job, as well as for a job they worked the longest in. Completely retired participants were also asked about their work schedule for their longest job ever worked, as well as the job they had before retirement. Response options for work schedules included: “daytime schedule or shift”, “evening shift”, “night shift”, “rotating shift, changing periodically from days to evenings or nights”, “seasonal, on-call or casual, no pre-arranged schedules”, and “other”. A binary variable for shiftwork was generated by categorizing the response “daytime schedule or shift” as “non-shiftwork” and combining all other work schedule responses into one category of “shiftwork”. Our measure of shiftwork only included those who had worked a duration of at least one year or longer at their main job/last job before retirement, prior to completing the questionnaire [1]. To better reflect duration of exposure, the main analyses used a measure of shiftwork which categorized participants based on their longest job ever worked. A second measure of shiftwork based on a participant’s most recent job was used in sensitivity analysis. 

### 2.3. Cognitive Performance Variables

The cognitive domains of memory and executive functioning have been recommended as outcome measures for assessments of cognitive functioning, as these domains are the most applicable for daily activities [22].

The Rey Auditory Verbal Learning Test (RAVLT), which assesses learning and retention [23], was used to measure declarative memory. Two of the RAVLT trials were administered by trained interviewers, with the first trial (immediate recall trial) involving a list of 15 words read out loud to participants, who were then tasked with immediately recalling the 15 words within 90 s, and a second trial (delayed recall trial) tasking participants to recall as many of the 15 words from the first trial within 60 s after a 30-min delay. One point was allocated for each correctly recalled word in each of the trials. Scores for the immediate recall trial and the delayed recall trial were used as continuous variables in the current analyses, with higher scores indicating greater cognitive performance. 

The Mental Alternation Test (MAT) [24] and the third subtask (i.e., “interference condition”) of the Stroop Test [25,26] were used to measure executive functioning. The MAT measures mental flexibility and processing speed [24]. In the CLSA, participants were asked to complete three subtasks, each within 30 s: (1) count from 1 to 20; (2) recite the alphabet out loud; (3) recite the alphabet in an alternating pattern with numbers (e.g., 1, A, 2, B, 3, C,…). Each correct alternation was allocated one point. The Stroop Test, which also consists of three sub-tasks, assesses inhibition, attention, mental speed, and mental control [25,26]. First, participants are presented with colored dots printed on cards and are tasked with identifying the color of each dot (Stroop 1) [23]. Another set of cards were presented with words printed in different colored ink and participants must name the ink colors of each word (Stroop 2) [23]. In the last subtask (interference condition), a set of cards which have color words (e.g., blue, green, red, yellow) printed in non-corresponding colored ink are presented and participants were asked to quickly name the color of the ink (e.g., say “blue” for the word “green” written in blue ink) [23]. Scores for the Stroop Test were based on how fast (in seconds) the task was completed [23]. For the current analyses, we only included the interference condition as a measure of executive functioning, as poorer performance on this part of the Stroop Test has been found among people with cognitive decline [23]. Scores for both the MAT and interference condition were used as continuous variables in the analyses. Higher scores for the MAT and shorter times recorded in the interference condition indicated better cognitive performance.

### 2.4. Moderator Variables

Psychological distress was assessed using the 10-item Kessler Psychological Distress Scale (K10), which measures non-specific distress [27]. An overall K10 score was obtained by summing responses to questions related to anxiety and depressive symptoms experienced in the previous month [27]. Response values ranged from 1 (“none of the time”) to 5 (“all of the time”) and total scores ranged from 10 to 50, with higher scores indicating greater levels of psychological distress. Psychological distress scores were dichotomized, with scores less than 20 indicating “low distress” and scores greater than or equal to 20 indicating “high distress” [28].

We stratified all analyses by retirement status within each sex to explore moderation by psychological distress. A binary measure for sex (male/female) was available in the CLSA. Retirement status was assessed at baseline by asking participants whether they considered themselves “completely retired”, “partly retired”, or “not retired”. The “partly retired” group was combined with the “not retired” group due to the limited number of observations in the “partly retired” category, thus forming two groups: completely retired and not/partly retired.

### 2.5. Covariates

The existing literature for shiftwork and cognitive impairment was used to identify potential confounders [9,29]. We adjusted for the following sociodemographic factors, which were measured at baseline: age (45–54, 55–64, 65–74, 75–85 years), education (less than secondary school, secondary school, some post-secondary, post-secondary degree/diploma), marital status (single/never married/never lived with a partner, married/common law, widowed/divorced/separated), household income (less than CAD 20,000, CAD 20,000–50,000, CAD 50,000–100,000, CAD 100,000–150,000, or CAD 150,000 or more), migrant status (non-immigrant/immigrant), place of residence (rural, urban, suburban), and social isolation (not socially isolated/socially isolated) [30].

Lifestyle factors controlled for in the analyses included smoking status (never, former occasional/daily, occasional/daily) and alcohol consumption (never, former, infrequent, occasional, regular, binge). Physical activity was measured using the self-reported Physical Activity Scale for the Elderly (PASE), which assesses different types of physical activities within the past week, including: walking outside; light, moderate, and strenuous sports or recreational activities; exercises to increase muscle strength or endurance; light and heavy housework or chores; home repairs, lawn, or yard maintenance; outdoor gardening; work or volunteer-related physical activity; and physical activity related to caring for other people [31]. Fruit and vegetable intake (seven or more, six, five, four, three, two, less than two servings per day) was measured using one item from the abbreviated version of the Seniors in the Community: Risk Evaluation for Eating and Nutrition Version II (SCREEN II) assessment tool [32].

We adjusted for general health (self-rated health: excellent or good/fair or poor), BMI categories (underweight/normal, overweight, obese), multimorbidity (0–1/≥2 chronic disease), and sleep quality (good/poor). Our measure of multimorbidity was based on the public health definition of multimorbidity [33], with the following chronic conditions: anxiety or mood disorder, arthritis, asthma, cancer, chronic obstructive pulmonary disease, diabetes, cardiovascular disease, and stroke. These chronic conditions were measured in the CLSA using the self-reported question, “Has a doctor ever told you that you have…?” Sleep satisfaction, also referred as subjective perception of sleep quality, is a measure of sleep health that has been found to be associated with mortality, metabolic syndrome, diabetes, high blood pressure, coronary heart disease, and depression [34]. CLSA participants were asked to self-report their level of satisfaction regarding their current sleep pattern. Participants who were “very dissatisfied” or “dissatisfied” with their current sleep pattern were categorized as having “poor sleep quality”, whereas those who were “neutral”, “satisfied” or “very satisfied” were considered as having “good sleep quality” [35].

### 2.6. Statistical Analyses

Descriptive analyses were stratified by retirement status within each sex. To examine the association between shiftwork and cognitive performance, unadjusted and adjusted linear regression were fitted separately for each outcome and stratified by retirement status within each sex [36]. All analyses used weights to account for the complex sampling design of the CLSA. To assess the potential moderating role of psychological distress, interaction terms were included in the linear regressions. Both crude and adjusted analyses were presented for psychological distress. 

Multiple imputation by chained equations was used to account for missing data for our main analysis (n = 5846/22,485, 26%). We specified 25 copies in the multiple imputation model and auxiliary variables (self-rated mental health, personal income, CES-D-10 depression score) were included to improve imputations. To assess the robustness of our findings, sensitivity analyses were performed that compared a complete case analysis to our imputed data, as well as our two shiftwork measures. Results from both sensitivity analyses were consistent with the main analysis (data not shown). All analyses were conducted using Stata/SE, version 16.1.

## 3. Results

### 3.1. Sample Characteristics

Of the 30,097 CLSA participants from the Comprehensive cohort, 22,485 participants met our inclusion criteria. As we excluded participants with missing data on cognitive outcome measures, our multivariable analyses included 20,610 participants in total. An overview of the sample of CLSA participants selected in this study, as well as the reasons for exclusion, is provided in Figure 1.

Characteristics of the CLSA participants who met the study inclusion criteria stratified by retirement status within each sex are presented in online Appendix A. The proportion of shift workers across all groups ranged from 14.5% to 20.1%. Most completely retired participants were between the ages of 65 to 74 years old, whereas those who were not/partly retired were under the age of 55. Most participants had attained post-secondary education, were former smokers, regular drinkers, overweight, reported having 0 to 1 chronic disease, had low psychological distress, and good sleep quality. Sample characteristics among shift workers by sex are shown in online Appendix A.

### 3.2. Shiftwork and Cognitive Performance

A weighted descriptive analysis of cognition scores stratified by retirement status within each sex is presented in Table 1. Female participants who were not/partly retired showed the highest scores for the immediate recall trial (mean = 6.5, SD: 1.7), delayed recall trial (mean = 5.0, SD: 2.1), and the interference condition (mean = 14.5, SD: 3.7), compared to participants in the other groups. Not/partly retired males showed the highest score for the MAT, with an average score of 28.7 (SD: 8.8). Cognition scores among shift workers by sex can be found in online Appendix A.

In Table 2, the estimated effect of shiftwork was attenuated for all cognitive measures after controlling for sociodemographic factors, lifestyle factors, general health, and chronic diseases. Associations which remained statistically significant were performances on the MAT (ß = −0.85, 95% CI: −1.21 to −0.50) and the interference condition (ß = 0.47, 95% CI: 0.31 to 0.63). No significant results were found for performances on the immediate (ß = −0.04, 95% CI: −0.11 to 0.04) and delayed recall trials (ß = −0.06, 95% CI: −0.14 to 0.03). 

In Table 3, stratifying by retirement status within each sex suggests that performance for executive functioning was the worst for both completely retired (MAT: ß = −1.10, 95% CI: −1.79 to −0.41; interference condition: ß = 0.58, 95% CI: 0.16 to 1.00) and not/partly retired males (MAT: ß = −1.10, 95% CI: −1.72 to −0.48; interference condition: ß = 0.59, 95% CI: 0.32 to 0.86). Of the female groups, only completed retired females who used to engage in shiftwork showed significantly poorer average scores on the MAT (ß = −0.82, 95% CI: −1.62 to −0.03). For both measures of declarative memory, no significant results were found for any of the stratified analyses (Table 3).

### 3.3. Psychological Distress on Shiftwork and Cognitive Performance

We assessed potential moderation by psychological distress on the relationship between shiftwork and cognitive performance. For all cognitive outcomes, there was a lack of significant moderating effect by psychological distress (online Appendix A). There was also no significant evidence of confounding by psychological distress (Table 2).

## 4. Discussion

In this very large, nationally representative sample, we found that shift workers showed poorer cognitive scores on tests for executive functioning but not for declarative memory, compared to non-shift workers. To our knowledge, this is the first Canadian study confirming this association in a very large, representative, population-based sample using objective cognitive measures. Given that cognitive processes are regulated by the circadian rhythm [6,7], misaligned circadian rhythm may represent one of the potential mechanisms explaining this relationship. As shiftwork takes place outside regular daytime working hours, working against the natural sleep–wake cycle may disrupt the circadian rhythm, resulting in impaired cognition in the long term [6,7]. Although our results align with prior population-based studies which have shown poor performance on tests for executive functioning among shift workers [10], our adjusted results for declarative memory contrast with other studies [8,11], as we did not find a significant association. Shiftwork may not impact all cognitive domains equally [37]. For example, findings from a randomized cross-over trial simulating shiftwork conditions found a steady performance on declarative memory during conditions of circadian rhythm alignment and misalignment [37]. This suggests that declarative memory performance may not be largely altered by circadian misalignment, in contrast to other cognitive domains such as processing speed and sustained attention [37].

We also investigated whether there were differences by sex and retirement status in this relationship. Males in both retirement groups showed poorer performance on executive functioning. Poorer scores on executive functioning were also found among completely retired females who used to engage in shiftwork, whereas no significant differences were observed for not or partly retired females. These findings suggest that the effects of shiftwork on cognition may be persistent as they are not reversed among those in retirement. Age has been consistently identified as the strongest risk factor for cognitive impairment [29]. As retired participants are likely older than non-retired participants, a reversal of effects in retirement may not be expected. However, a prospective cohort study by Bokenberger and colleagues [13] demonstrated no significant association between shiftwork and cognition at retirement age. Furthermore, prospective cohort [8] and cross-sectional studies [10] have found no significant differences between non-shift workers and those who have left shiftwork for more than five years. We were unable to explore this in our study, as we did not have information on the length of time since leaving shiftwork among our sample. 

Engaging in shiftwork may elicit stress and lead to psychological distress [16]. People who are more susceptible to psychological distress may experience greater rates of cognitive decline compared to those less prone [38]. Furthermore, psychological distress may be present in people with mild cognitive impairment and increase the risk of dementia progression [39]. Our findings suggest that the magnitude of the association between shiftwork and cognitive functioning may be the same regardless of the level of psychological distress. To the best of our knowledge, no other study has examined the potential moderating role of psychological distress on the relationship between shiftwork and cognitive performance. Future longitudinal studies are warranted to further explore this relationship and confirm this finding.

Our study had several strengths and limitations. Major strengths of our study include the very large, nationally representative population-based sample, as well as the inclusion of a number of relevant covariates and gold standard measures for our cognitive outcomes. For limitations, occupation type was not examined and the association between shiftwork and cognitive performance may not be constant across all jobs. Due to limited observations, we were unable to separate the effects of the different types of shiftwork schedules on cognition [8]. Misclassification may be present in our study due to discordance in work schedules between the jobs included in our shiftwork definition, as well as the self-reported assessment of sleep quality. Our study may contain selection bias as the CLSA Comprehensive cohort only recruited people living within 25 to 50 km of data collection sites in 7 of the 10 provinces in Canada [20]. Moreover, most participants included in this cohort are non-immigrants, have better self-rated health, are more educated, and have higher household incomes [20]. The cross-sectional design of our study is a major limitation precluding us from inferring temporality and directionality on the relationship between shiftwork and cognitive performance, with the possibility of reverse causation. Finally, future longitudinal studies are needed to confirm the observed associations and determine the clinical meaning of our results.

## 5. Conclusions

The use of shiftwork schedules will continue to sustain the continuous operation of goods and services. Our study provides suggestive evidence of a potential association between shiftwork and cognitive performance, particularly in the domain of executive functioning, but not for declarative memory. Our findings may help inform employers and health and safety policy committees to better design shiftwork schedules that are less disruptive to the circadian rhythm [40]. Males and females showing lower cognitive scores for executive functioning may help inform employees about potential risks involved in working in the shift system and consider the impact it may have on daily activities which rely on sound cognitive functioning [8]. Although we did not find a significant moderating effect by psychological distress, employers and health and safety policy committees should still create workplace environments aimed at promoting mental well-being that will help people cope with shift schedules [40]. Findings of this study contribute to the literature on shiftwork and cognitive performance, especially in the Canadian context. Future research using a prospective cohort design is warranted.

## Figures and Tables

**Figure 1 ijerph-19-10124-f001:**
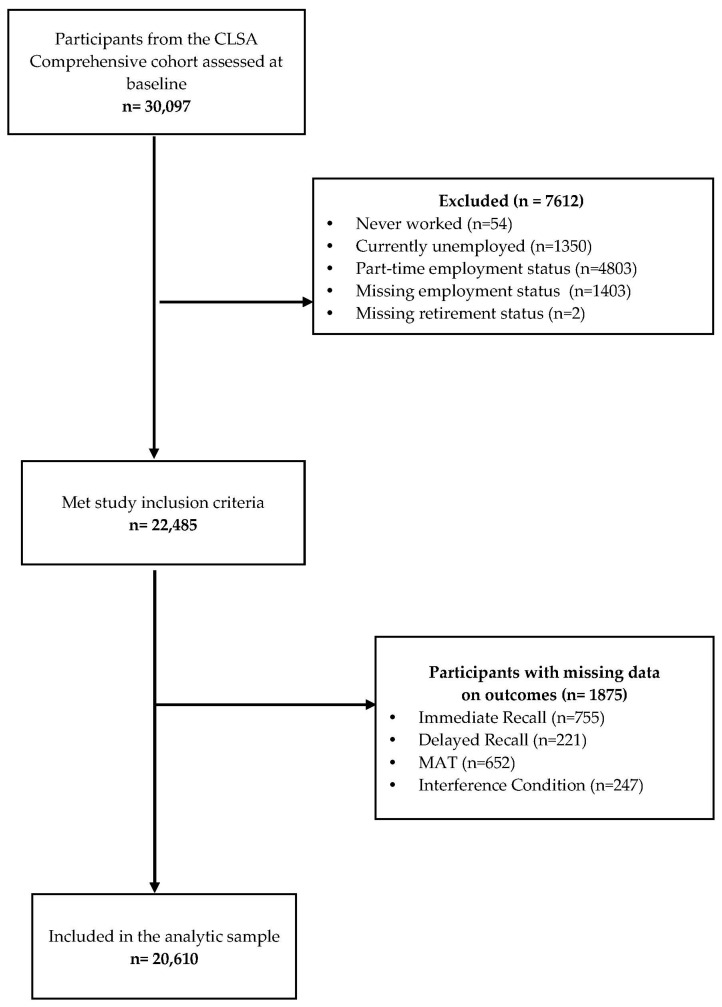
Flow chart outlining inclusion and exclusion of CLSA participants.

**Table 1 ijerph-19-10124-t001:** Weighted descriptive analysis of cognition scores, stratified by retirement status and sex (N = 20,610).

Cognition Scores	All (N = 20,610)	Males		Females	
		Completely Retired (n = 4884)	Not/Partly Retired (n = 6374)	Completely Retired (n = 4708)	Not/Partly Retired (n = 4644)
	n	n	n	n	n
**Immediate Recall (0–14 points)**					
Mean (SD)	5.8 (1.9)	4.8 (2.0)	5.7 (1.6)	5.6 (1.9)	6.5 (1.7)
**Delayed Recall (0–14 points)**					
Mean (SD)	4.0 (2.1)	2.9 (2.1)	4.0 (1.8)	3.8 (2.2)	5.0 (2.0)
**MAT Score (0–51 points)**					
Mean (SD)	26.6 (9.0)	24.4 (10.4)	28.7 (8.2)	23.5 (9.3)	27.5 (7.8)
**Interference Condition (1–132 s)**					
Mean (SD)	15.9 (4.7)	18.4 (6.2)	14.7 (3.4)	17.7 (5.8)	14.3 (3.1)

Acronyms: SD—standard deviation.

**Table 2 ijerph-19-10124-t002:** Association between shiftwork and performance on cognitive tests for declarative memory and executive functioning among CLSA participants.

	Unadjusted Association		Adjusted ^a^ Association		Fully Adjusted ^b^ Association	
	ß	95% CI	ß	95% CI	ß	95% CI
**Immediate Recall Trial**						
Shiftwork (Yes vs. No)	**−0.19**	**−0.27, −0.11**	−0.04	−0.11, 0.04	−0.04	−0.11, 0.04
**Delayed Recall Trial**						
Shiftwork (Yes vs. No)	**−0.20**	**−0.29, −0.11**	−0.06	−0.14, 0.03	−0.06	−0.14, 0.03
**MAT**						
Shiftwork (Yes vs. No)	**−1.46**	**−1.84, −1.09**	**−0.85**	**−1.21, −0.50**	**−0.85**	**−1.21, −0.50**
**Interference Condition**						
Shiftwork (Yes vs. No)	**0.79**	**0.61, 0.98**	**0.47**	**0.31, 0.64**	**0.47**	**0.31, 0.64**

Acronyms: CI—confidence interval. Significant results are bolded. ^a^: adjusted for sociodemographic factors, lifestyle factors, general health, and chronic diseases. ^b^: adjusted for psychological distress and confounders previously controlled for in the adjusted model.

**Table 3 ijerph-19-10124-t003:** Multiple linear regression models for shiftwork and performance on cognitive tests, stratified by retirement status within each sex.

	**Males**		**Females**	
	**Completely Retired** **(n = 4884)**	**Not/Partly Retired** **(n = 6374)**	**Completely Retired** **(n = 4708)**	**Not/Partly Retired** **(n = 4644)**
**Immediate Recall Trial**				
Model 1 ß (95% CI)	**−0.25 (−0.39, −0.11)**	**−0.20 (−0.33, −0.07)**	−0.17 (−0.34, 0.00)	−0.04 (−0.21, 0.13)
Model 2 ß (95% CI)	−0.10 (−0.23, 0.03)	−0.06 (−0.18, 0.07)	−0.05 (−0.21, 0.11)	0.06 (−0.10, 0.23)
**Delayed Recall Trial**				
Model 1 ß (95% CI)	**−0.19 (−0.34, −0.04)**	**−0.19 (−0.34, −0.05)**	**−0.21 (−0.39, −0.03)**	−0.10 (−0.29, 0.10)
Model 2 ß (95% CI)	−0.06 (−0.20, 0.08)	−0.05 (−0.19, 0.09)	−0.12 (−0.29, 0.05)	−0.02 (−0.21, 0.17)
	**Males**		**Females**	
	**Completely Retired** **(n = 5484)**	**Not/Partly Retired** **(n = 6886)**	**Completely Retired** **(n = 5149)**	**Not/Partly Retired** **(n = 4966)**
**MAT**				
Model 1 ß (95% CI)	**−1.95 (−2.68, −1.23)**	**−1.88 (−2.52, −1.23)**	**−1.57 (−2.42, −0.72)**	**−0.75 (−1.46, −0.05)**
Model 2 ß (95% CI)	**−1.10 (−1.79, −0.41)**	**−1.10 (−1.72, −0.48)**	**−0.82 (−1.62, −0.03)**	−0.29 (−0.98, 0.41)
**Interference Condition**				
Model 1 ß (95% CI)	**1.08 (0.62, 1.54)**	**0.93 (0.65, 1.21)**	**0.75 (0.31, 1.19)**	**0.44 (0.13, 0.75)**
Model 2 ß (95% CI)	**0.58 (0.16, 1.00)**	**0.59 (0.32, 0.86)**	0.31 (−0.10, 0.72)	0.25 (−0.04, 0.54)

Acronyms: CI—confidence interval. Significant results are bolded. Model 1: unadjusted association between shiftwork and performance on cognitive tests. Model 2: adjusted association between shiftwork and performance on cognitive tests.

## Data Availability

Data are available from the Canadian Longitudinal Study on Aging (www.clsa-elcv.ca; accessed on 9 August 2021) for researchers who meet the criteria for access to de-identified CLSA data.

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
