# Peer review of "Does Shiftwork Impact Cognitive Performance? Findings from the Canadian Longitudinal Study on Aging (CLSA)"

_ijerph, 2022, doi:10.3390/ijerph191610124_

Round 1

Reviewer 1 Report

The authors examined whether and how shiftwork was associated with cognitive performance using the data collected from the Comprehensive Cohort of the Canadian Longitudinal Study on Aging (CLSA). The article was well written and was easy to follow. The paper can be published after minor revisions.

1st Comment: Abstract, lines 8-9 (and Page 5, lines 204-205): “beta=0.47, 95% CI: 0.21 to 0.63” should be “beta=0.47, 95% CI: 0.31 to 0.64”. [Please check your data shown in the last row of Table 2 on page 5].

2nd Comment: Figure 1. “Excluded (n = 7,610)” should be “Excluded (n = 7,612)” because it should be the sum of 54, 1350, 4803, 1403, and 2.

3rd Comment: In Abstract (lines 5-6), the authors stated that “… among 20,610 community-dwelling adults…” The authors also indicated (lines 178-180) that “…22,485 participants met our inclusion criteria. As we excluded participants with missing data on cognitive outcome measures, our multivariable analyses included 20,610 participants in total.” Additionally, Figure 1 shows “Included in the analytic sample n = 20,610.” However, when I looked at Tables 1 and 3, the number of respondents was 22,485 (for example in Table 3, 5484+6886+5149+4966=22,485). In order not to confuse readers about the number of respondents included in the final analysis, I suggest the authors to update Tables 1 and 3 accordingly (i.e. just presenting the number of respondents (and total number of respondents) who answered all items of cognitive performance.   

4th Comment: The page number of Ref. 2 (i.e. Almondes and Araujo, 2009) should be “15-23.”

Author Response

Response to Reviewer 1 Comments

Point 1: Abstract, lines 8-9 (and Page 5, lines 204-205): “beta=0.47, 95% CI: 0.21 to 0.63” should be “beta=0.47, 95% CI: 0.31 to 0.64”. [Please check your data shown in the last row of Table 2 on page 5]

Response 1: We greatly thank you for pointing out this typo. We have double checked the data and have included the correct confidence interval in the abstract and in the results section.

Point 2: Figure 1. “Excluded (n = 7,610)” should be “Excluded (n = 7,612)” because it should be the sum of 54, 1350, 4803, 1403, and 2

Response 2: We appreciate you finding this error in our figure. We have changed the total number for participants excluded to 7,612 in Figure 1.

Point 3: In Abstract (lines 5-6), the authors stated that “… among 20,610 community-dwelling adults…” The authors also indicated (lines 178-180) that “…22,485 participants met our inclusion criteria. As we excluded participants with missing data on cognitive outcome measures, our multivariable analyses included 20,610 participants in total.” Additionally, Figure 1 shows “Included in the analytic sample n = 20,610.” However, when I looked at Tables 1 and 3, the number of respondents was 22,485 (for example in Table 3, 5484+6886+5149+4966=22,485). In order not to confuse readers about the number of respondents included in the final analysis, I suggest the authors to update Tables 1 and 3 accordingly (i.e. just presenting the number of respondents (and total number of respondents) who answered all items of cognitive performance.

Response 3: We thank the reviewer for their suggestion. We have updated the number of respondents in Table 1 and 3 and have provided updated numbers for the cognition scores in Table 1.

Point 4: The page number of Ref. 2 (i.e. Almondes and Araujo, 2009) should be “15-23.”

Response 4: We have added these page numbers into reference. We sincerely thank you for providing the page numbers.

Reviewer 2 Report

The manuscript by Rea Alonzo et al., entitled:

Does Shiftwork Impact Cognitive Performance? Findings from the Canadian Longitudinal Study on Aging (CLSA)”

addresses the tentative correlation between cognitive performance (declarative memory and executive functioning) in shiftworkers, by statistically analyzing differences of the 17.2% shiftworkers out of the 22,485 adults from the Comprehensive Cohort of the Canadian Longitudinal Study on Aging (CLSA), aged 45-85 years old at enrollment, who met the inclusion criteria.  

Authors analyzed differences of shiftworkers on Rey Auditory Verbal Learning Test (RAVLT) to measure declarative memory, and on Mental Alternation Test (MAT) and the third sub-task (i.e. “interference condition”) of the Stroop Test to measure executive functioning.

Associations with psychological distress were also explored by using the 10-item Kessler Psychological Distress Scale (K10), to better understand whether psychological distress may play a moderating role between shiftwork and cognitive performance.

Results were adjusted by potential confounders such as sociodemographic factors (age, educations marital status, household income, migrant status, place of residence and social isolation), lifestyle factors and general health.

Differences were reported by sex and retirement status.

Authors found that shiftwork was significantly associated with poorer performance for executive functioning but not for declarative memory, however, no evidence of moderating effect by psychological distress was found.

Authors cleverly exploited previous data to investigate shiftwork impact on cognitive performance. The manuscript is well-structured, well-written and adds up to literature aiming to better understand cognitive performance in shift workers.

Thus I recommend it for publications.

However, a major comment should be considered:

Authors show a number of tables and supplementary data in which it is not possible to see the original data (or percentages) of shifworkers in considered variables and covariates. I.e. authors just show data stratified by status and sex (e.g. Table 1 and Suppl. Table 1), but not by shiftworkers. Otherwise, authors just show statistical results (e.g. Table 2).

Thus, it is strongly recommended to include, maybe as Supplementary table 2, a table devoted to the 17.2% shiftworkers similar to Supp. Table 1, detailing age, education, etc. of the shifworkers.

It could be detailed by sex.

Additionally, another table should show detailed results of the cognition scores of the shiftworkers.

It could also be detailed by sex.

Other minor comments follow:

-          Page 1: Please, check bold letter and spaces after Abstract.

-          Page 7: Please, check bold letter and spaces after Acknowledgements, Disclosure statement, data availability statement and Author Contributions.

-          Supplementary table 1. You may want to adjust numbers in columns. For instance, in BMI, when adding up all the first line the total result is 100.4% (instead of exactly 100%, as should be expected). This also happens in other sections of the table.

Author Response

Response to Reviewer 2 Comments

Point 1: Authors show a number of tables and supplementary data in which it is not possible to see the original data (or percentages) of shifworkers in considered variables and covariates. I.e. authors just show data stratified by status and sex (e.g. Table 1 and Suppl. Table 1), but not by shiftworkers. Otherwise, authors just show statistical results (e.g. Table 2). Thus, it is strongly recommended to include, maybe as Supplementary table 2, a table devoted to the 17.2% shiftworkers similar to Supp. Table 1, detailing age, education, etc. of the shifworkers.

It could be detailed by sex.

Response 1: We appreciate the reviewer’s suggestion and have included an additional table in the supplementary (supplemental table 2) showing characteristics of shiftworkers by sex.

Point 2: Additionally, another table should show detailed results of the cognition scores of the shiftworkers.

It could also be detailed by sex.

Response 2: We agree with the reviewer’s suggestion and have included another table in the online supplementary (supplemental table 3) showing cognition scores of shiftworkers by sex.

Point 3: Page 1: Please, check bold letter and spaces after Abstract.

Response 3: We greatly thank the reviewer for pointing this out. We have fixed the bolded word in the abstract.

Point 4: Page 7: Please, check bold letter and spaces after Acknowledgements, Disclosure statement, data availability statement and Author Contributions.

Response 4: We have fixed the bolded letters in the Acknowledgements, Disclosure statement, data availability statement and Author Contributions. We thank the reviewer for finding these typos in the manuscript.

Point 5: Supplementary table 1. You may want to adjust numbers in columns. For instance, in BMI, when adding up all the first line the total result is 100.4% (instead of exactly 100%, as should be expected). This also happens in other sections of the table.

Response 5: We appreciate the reviewer’s suggestion to adjust the numbers in Supplementary table 1. We have reviewed our data again and have adjusted our numbers in the table. As numbers in the table have been rounded, some column may not add up to exactly 100%.

Reviewer 3 Report

  Comments to the Author:

This manuscript presents data on the relationship of shiftwork and cognitive performance. Given the impact of shiftwork on sleep quality, and the importance of sleep for health and quality of life, the present study is of high importance. However, I have several, mainly methodological issues that need to be addressed:

1.       Line 18 – “conflicting findings” – only one study is discussed, better to present opposite finding as well.

2.       Lines 55-56 - Participants who have never worked or who were currently unemployed at the time of the survey were also excluded” What about completely retired people? Were they employed at the time of survey?

3.       Lines 62-63 – for completely retired participants, questions were asked only about longest job and a before retirement job? What about the time? How long they are retired? how long they are without a job?

4.       Lines 64-70 - Response options for work schedules are unclear - “daytime schedule or shift” – how you know what kind of shift person has, why this response means “non-shiftwork”, -absolutely not clear. Further – response option “other” why you consider it as a shiftwork or how you know when and how long the work was performed? These issues are very important in order to discuss obtained results.

5.       Line 70 – “who have worked a duration of at least one year or longer at their main job/last job before retirement, prior to completing the questionnaire” – do you mean that those subjects were retired recently, just before assessment?

6.       Line 132 – how social isolation was measured?

7.       Line 153 – Better to formulate as “sleep satisfaction also referred as subjective perception of sleep quality”

8.       Table 1 – “Missing” values– describes number of participants, right?  Better add “n” to “Missing” as “Missing (n)”, then on the right side.

9.       Table 2 – All beta and 95% CI values for all variables for adjusted and fully adjusted associations are absolutely the same?

10.   Line 211 – what are other confounders should be described

11.    Discussion, second paragraph, second line, “both retirement groups” – since you combined partly retired and not-retired, there is no 2 retirement groups.

12.   Results, section 3.3 - moderation by psychological distress was one of the aim of your study, and data should be presented clearly. You refer to table 2 for moderation but along with psychological distress you have “other variables” which may affect the results of confounding by psychological distress. Further, as already noted, to have absolutely the same values for adjusted and fully adjusted associations is unlikely.

13.   No study limitations are present.

Author Response

Response to Reviewer 3 Comments

Point 1: Line 18 – “conflicting findings” – only one study is discussed, better to present opposite finding as well.

Response 1: Four population-based studies which have examined the chronic effects of shiftwork and present conflicting findings across each other have been cited in line 18. The study by Wong et al is a Canadian study and is discussed in the manuscript to demonstrate the lack of Canadian studies examining shiftwork and cognitive function. We thank the reviewer for their comment and have updated the first sentence of this paragraph (line 12-14) to clarify this.

Point 2:  Lines 55-56 - Participants who have never worked or who were currently unemployed at the time of the survey were also excluded” What about completely retired people? Were they employed at the time of survey?

Response 2: Participants in the CLSA were given separate sets of questions related to employment based on whether they self-reported being “not retired”, “partly retired”, or “completely retired”. Participants who self-reported being “not retired” were asked whether they have never worked or were currently unemployed at the time of the survey and were excluded from the sample accordingly. Participants who self-reported as being “completely retired” were not employed at the time of the survey and were not asked about their current employment status.

Point 3:    Lines 62-63 – for completely retired participants, questions were asked only about longest job and a before retirement job? What about the time? How long they are retired? how long they are without a job?

Response 3: Yes, that is correct. The CLSA only asked completely retired participants questions pertaining to their longest job and job before retirement. Questions about the length of time participants have been in retirement were not provided by the CLSA.

Point 4:     Lines 64-70 - Response options for work schedules are unclear - “daytime schedule or shift” – how you know what kind of shift person has, why this response means “non-shiftwork”, -absolutely not clear. Further – response option “other” why you consider it as a shiftwork or how you know when and how long the work was performed? These issues are very important in order to discuss obtained results.

Response 4: In the employment-related questionnaires distributed by the CLSA, participants were asked to report their type of work schedule by choosing one of the following response options: “daytime schedule or shift”, “evening shift”, “night shift”, “rotating shift, changing periodically from days to evenings or nights”, “seasonal, on-call or casual, no pre-arranged schedules” and “other”. As shiftwork is defined as atypical work schedules occurring outside regular business hours, we defined “daytime schedule or shift” as “non-shiftwork” and included “other” as “shiftwork”. Given the lack of data availability from the CLSA, we were unable to determine the length of time of each work schedule.

Point 5:         Line 70 – “who have worked a duration of at least one year or longer at their main job/last job before retirement, prior to completing the questionnaire” – do you mean that those subjects were retired recently, just before assessment?

Response 5: Yes, that is correct. In the employment-related questionnaires, participants who self-reported being “not retired” or “partly retired” were asked to report the length of time in years worked at their main job, while participants who self-reported being “completely retired” were asked to report the length of time in years worked at their last job before entering retirement. For participants who were “not retired” or “partly retired”, only those who reported to have already worked at least 1 year in their main job were included in the sample. Among participants were who “completely retired”, those who reported to have worked a duration of at least 1 year in their last job before entering retirement were included.

Point 6:         Line 132 – how social isolation was measured?

Response 6: To measure social isolation, we followed the methods provided by a pervious CLSA study (Menec et al, 2019) which has been cited in our manuscript (line 134). First, five indicators of social isolation were created using questions from the CLSA measuring a person’s social network and participation in social activities. The social isolation index was generated by allocating one point to each indicator, resulting in a continuous variable ranging from zero to five. We then dichotomized the social isolation index variable, classifying people with scores of zero to two as “not socially isolated” and scores three to five as “socially isolated”.

Point 7:          Line 153 – Better to formulate as “sleep satisfaction also referred as subjective perception of sleep quality”

Response 7: We thank the reviewer for their suggestion and have made the change in the manuscript (line 154).

Point 8:       Table 1 – “Missing” values– describes number of participants, right?  Better add “n” to “Missing” as “Missing (n)”, then on the right side.

Response 8: We thank the reviewer’s suggestion. We have updated table 1 in the manuscript to show cognition scores among those with non-missing cognitive outcomes and have removed columns showing

Point 9:   Table 2 – All beta and 95% CI values for all variables for adjusted and fully adjusted associations are absolutely the same?

Response 9: The values presented in the adjusted and fully adjusted model appear the same as the final numbers have been rounded.

Point 10:    Line 211 – what are other confounders should be described

Response 10: The other confounders adjusted for in the fully adjusted model include sociodemographic factors, lifestyle factors, general health, and chronic diseases, which were also controlled for in the adjusted model. We have clarified this in the footnotes of table 2 (line 213).

Point 11: Discussion, second paragraph, second line, “both retirement groups” – since you combined partly retired and not-retired, there is no 2 retirement groups

Response 11: In the CLSA, retirement status was assessed at baseline by asking participants whether they considered themselves “completely retired”, “partly retired” or “not retired”. The “partly retired” group was combined with the “not retired” to form one group, leaving those “completely retired” as a separate group. We appreciate the reviewer’s comment and agree that our description of the two retirement groups was not clear in the methods. We have added an additional statement in the methods (lines 123-124) to clarify the two retirement groups.

Point 12: Results, section 3.3 - moderation by psychological distress was one of the aim of your study, and data should be presented clearly. You refer to table 2 for moderation but along with psychological distress you have “other variables” which may affect the results of confounding by psychological distress. Further, as already noted, to have absolutely the same values for adjusted and fully adjusted associations is unlikely.

Response 12: We thank the reviewer for their comment and will present the data more clearly. For the moderation results, we have included and referenced in the manuscript a supplementary table (Supplemental Material Table 4) which present the estimates of the interaction effect between shiftwork and psychological for each cognitive outcome. Values in the table appear to be the same due to rounding.

Point 13:   No study limitations are present.

Response 13: There were several limitations present in our study. First, we were unable to separate the effects of the different types of shiftwork schedules on cognition as well as examine the occupation type of our participants. As our measure of sleep quality was self-reported, and due to differences in work schedules between jobs included in our definition of shift work, it is possible for misclassification in our study. Selection bias may also be present due to the nature of recruitment of participants in the Comprehensive cohort by the CLSA. Finally, the cross-sectional design of our study presents another limitation. These study limitations are discussed in more detail in the fourth paragraph of the Discussion section.

References

Menec VH, Newall NE, Mackenzie CS, et al. Examining individual and geographic factors associated with social isolation and loneliness using Canadian Longitudinal Study on Aging (CLSA) data PLoS One 2019;14(2):e0211143.
